# Allelic variants of full-length VAR2CSA, the placental malaria vaccine candidate, differ in antigenicity and receptor binding affinity

Jonathan P. Renn[1], Justin Y. A. Doritchamou [1], Bergeline C. Nguemwo Tentokam[1], Robert D. Morrison[1], Matthew V. Cowles[1], Martin Burkhardt[1], Rui Ma [1], Almahamoudou Mahamar[2], Oumar Attaher[2], Bacary S. Diarra[2], Moussa Traore[2], Alassane Dicko[2], Niraj H. Tolia [1], Michal Fried[1] & Patrick E. Duffy [1✉]

*Plasmodium falciparum*-infected erythrocytes (IE) sequester in the placenta via surface protein VAR2CSA, which binds chondroitin sulfate A (CSA) expressed on the syncytiotrophoblast surface, causing placental malaria (PM) and severe adverse outcomes in mothers and their offspring. VAR2CSA belongs to the PfEMP1 variant surface antigen family; PfEMP1 proteins mediate IE adhesion and facilitate parasite immunoevasion through antigenic variation. Here we produced deglycosylated (native-like) and glycosylated versions of seven recombinant full-length VAR2CSA ectodomains and compared them for antigenicity and adhesiveness. All VAR2CSA recombinants bound CSA with nanomolar affinity, and plasma from Malian pregnant women demonstrated antigen-specific reactivity that increased with gravidity and trimester. However, allelic and glycosylation variants differed in their affinity to CSA and their serum reactivities. Deglycosylated proteins (native-like) showed higher CSA affinity than glycosylated proteins for all variants except NF54. Further, the gravidity-related increase in serum VAR2CSA reactivity (correlates with acquisition of protective immunity) was absent with the deglycosylated form of atypical M200101 VAR2CSA with an extended C-terminal region. Our findings indicate significant inter-allelic differences in adhesion and seroreactivity that may contribute to the heterogeneity of clinical presentations, which could have implications for vaccine design.

[1] Laboratory of Malaria Immunology and Vaccinology, National Institute of Allergy and Infectious Diseases, National Institutes of Health, Bethesda, MD, USA.
[2] Malaria Research and Training Center, University of Sciences, Techniques, and Technologies of Bamako, Bamako, Mali. ✉email: patrick.duffy@nih.gov

Placental malaria (PM) can cause poor outcomes for pregnant women and their babies, including severe maternal anemia, low birth weight delivery and fetal loss[1]. In Africa, PM has been estimated to cause up to 50,000 maternal deaths, ~200,000 infant deaths, and nearly a third of perinatal mortality in the absence of preventive measures[1,2]. In malaria-endemic areas, the immunity that protects individuals from malaria is thought to be progressively acquired from childhood to adult age[3,4]. Despite this pre-existing immunity, women become susceptible to *P. falciparum* parasitemia during pregnancy leading to PM, especially during first gestation[5]. Immunity to PM is acquired over successive pregnancies[6].

PM is characterized by the sequestration of *P. falciparum*-infected erythrocytes (IE) in placental intervillous spaces, which elicits inflammatory infiltrates and cytokine responses associated with poor outcomes[7]. Placental sequestration results from IE that express the variant surface antigen VAR2CSA on their surface and bind to Chondroitin Sulfate A (CSA) in intervillous spaces and on the syncytiotrophoblast surface[8,9]. VAR2CSA is a member of the *Plasmodium falciparum* erythrocyte membrane protein 1 (PfEMP1) family, which plays a central role in virulence by mediating adhesion and sequestration, as well as in immunoevasion through antigenic variation[10,11]. VAR2CSA is a large cysteine-rich transmembrane multidomain protein (~350 kDa) typically formed by six Duffy-binding-like (DBL) domains with several interdomain (ID) regions[12,13] although recent work has identified atypical VAR2CSA forms with seven or more DBL domains in maternal isolates of *P. falciparum*. Structural studies by cryo-electron microscopy show that the multiple domains of VAR2CSA interact extensively to create an interwoven architecture, and that CSA binds by threading through a major and minor channel of VAR2CSA formed by NTS, DBL1X, DBL2X, ID2α, and DBL4ε domains as part of a stable core (extending from NTS to ID3) that is flanked by a flexible DBL5-DBL6 arm[14].

VAR2CSA is the leading vaccine candidate to prevent PM based on multiple lines of evidence. Parasites isolated from the placenta uniformly express VAR2CSA[15,16]. High levels of antibodies that inhibit placental IE adhesion and antibodies to VAR2CSA are associated with improved outcomes of multigravidae[6,17]. Women who have naturally acquired anti-adhesion antibodies are protected against PM-related adverse pregnancy outcomes[17,18] and this serum anti-adhesion activity is globally cross-reactive[19]. Anti-adhesion activity has been induced by VAR2CSA vaccination in preclinical studies[20,21]. Finally, recombinant VAR2CSA (both full-length and smaller fragments) bind to CSA with nanomolar affinity[22]. These observations suggest that antibodies binding to VAR2CSA could block the interaction of VAR2CSA with CSA and thereby prevent placental sequestration of IE.

Unfortunately, VAR2CSA-based vaccine development has been stymied by the highly polymorphic sequence in VAR2CSA, its large protein size and abundance of cysteine residues that make the expression in systems such as *E. coli* or yeast infeasible for vaccine purposes. VAR2CSA-based vaccine design thus has relied on identifying functional fragments of VAR2CSA that bind CSA with high affinity and induce broadly neutralizing antibodies[23]. Two VAR2CSA-based vaccine candidates (PAMVAC and PRIMVAC) generated from N-terminal fragments of the protein are currently in phase I/II clinical trials[24,25]. First-in-human reports from these trials have indicated that PAMVAC and PRIMVAC vaccines are safe, well-tolerated and induce functional antibodies to homologous parasites in malaria-naive and malaria-exposed women[26,27]. However, vaccinees did not acquire strain-transcending serum functional activity[26,27], suggesting subunit vaccines based on VAR2CSA fragments may elicit a restricted breadth of functional antibodies in humans. As CSA

binds by threading through a major and minor channel of VAR2CSA, altogether formed by NTS, DBL1X, DBL2X, ID2α, and DBL4ε domains as part of a stable core (extending from NTS to ID3)[14], larger fragments or full-length VAR2CSA will more faithfully recapitulate the binding site and possibly critical epitopes of functional antibodies.

Although challenges remain to produce full-length VAR2CSA under good manufacturing practice (GMP) conditions, recent successes expressing a full-length VAR2CSA variant in mammalian[28–30] or insect cells[31] enable improved production to better characterize this protein biophysically, structurally, and immunologically. VAR2CSA recombinants generated from these two different expression systems bound well to CSA and induced potent anti-adhesion antibodies in rodents and rabbits, but without cross-inhibition activity against heterologous parasite strains[31,32]. Due to the high sequence diversity in VAR2CSA, recombinant forms of multiple alleles will be critical to characterize key epitopes that induce broadly neutralizing activity.

Here, we report successful high-yield expression and purification of recombinant full-length VAR2CSA proteins using sequences from seven different *P. falciparum* alleles in an optimized eukaryotic expression system. We demonstrate that recombinant full-length extracellular VAR2CSA antigens bind to CSA with nanomolar affinity and are recognized by naturally acquired antibodies from malaria-exposed individuals, albeit the levels of binding and reactivity vary between alleles. Our results establish biophysical, functional, and antigenic differences between diverse full-length VAR2CSA proteins.

## Results

**Selection of full-length VAR2CSA sequences for protein expression.** Seven different strains of *P. falciparum* were selected from distinct clusters of VAR2CSA sequences following phylogenetic analysis (Fig. 1a), and the frequency of amino acid mutations was analyzed across VAR2CSA extracellular domains (Fig. 1b). NF54 and FCR3 were selected as these alleles are currently under clinical evaluation as PM vaccine candidates, and distinct in their dimorphic sequence motif (DSM) in the DBL2X domain of VAR2CSA[33]. The 7G8, M. Camp and HB3 alleles were chosen as they fall in phylogenetic clusters distinct from NF54 and FCR3. The maternal isolate M200101 was included as it has an atypical structure with an additional DBL7ε domain downstream of DBL6ε[34]. Finally, the Tanzanian maternal isolate M920 sequence was selected as it has a different DSM in ID1[35] than the one present in all previously selected isolates. These selected alleles embody to some extent the sequence variation and geographic diversity of VAR2CSA (Table 1). Extracellular region sequences from the seven alleles were used to generate recombinant proteins.

**Recombinant full-length VAR2CSA variants are secreted in cell culture media and show biophysical differences.** Seven different full-length VAR2CSA proteins were expressed in mammalian cell line HEK293. Full-length VAR2CSA contains a high number of cysteine residues that can form disulfide bonds. In order to facilitate correct folding and disulfide bond formation, the full-length VAR2CSA protein sequence was cloned after the N-terminal signal peptide to guide the protein to the endoplasmic reticulum (ER) (Fig. 1c), which has many chaperones including machinery to form correct disulfide bonds[36]. In addition, vectorial appearance of the nascent polypeptide chain into the ER milieu can facilitate correct folding and disulfide bond formation[36]. A C-terminal his-tag was added to aid in purification and identification. HEK293F or Expi293 cells in culture volumes ranging from 0.5L-2L were transfected with

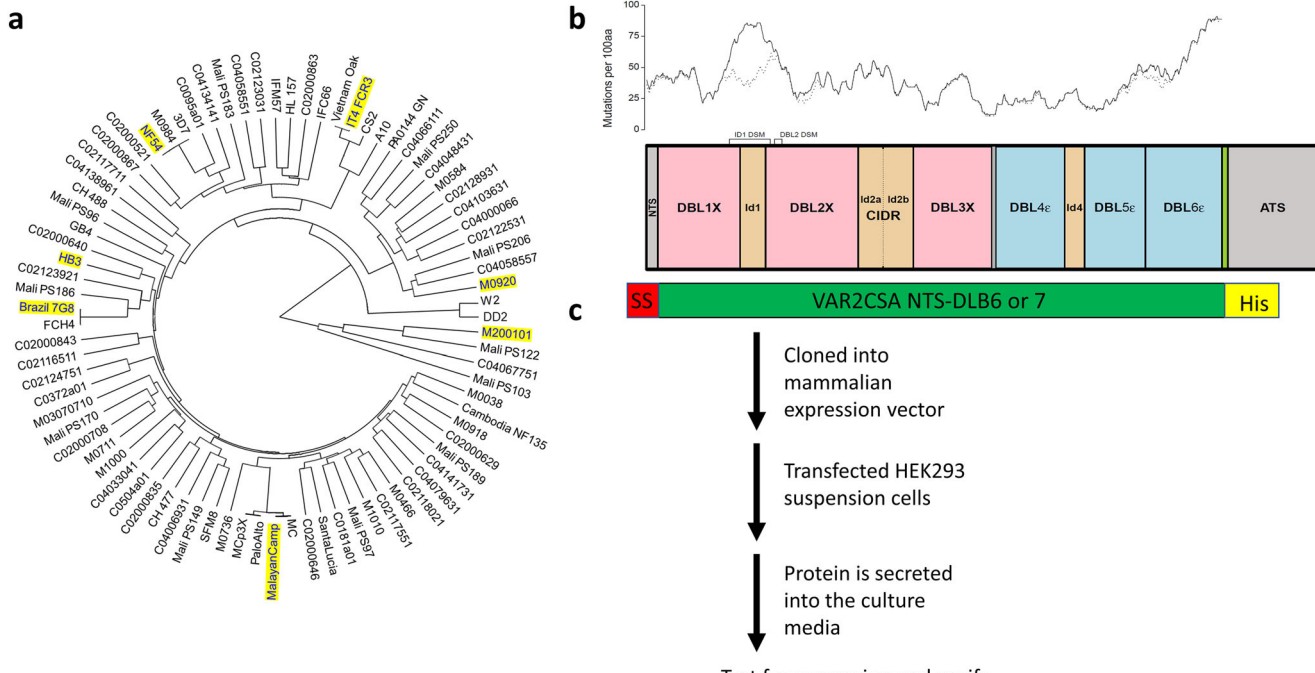

**Fig. 1 Phylogenetic analysis of VAR2CSA sequences from different isolates, frequency of amino acid mutations across VAR2CSA extracellular domains for the seven VAR2CSA alleles studied, and workflow of VAR2CSA expression. a** Phylogenetic tree showing full-length VAR2CSA amino acid sequences from lab strains and isolates from different geographic locations. The seven sequences selected for this study are highlighted in yellow. **b** Schematic organization of VAR2CSA protein for FCR3, drawn to scale, showing its N-terminal sequence, 6 cysteine-rich Duffy binding like (DBL) domains, inter-domain (ID) regions, transmembrane domain (TMD), and intracytoplasmic tail (ATS). Prevalence of sequence variability along the protein's extent, using DBL1-6 from the seven full-length constructs (DBL7 from M20010 is not included in this analysis), is graphically displayed as a solid black line showing the number of amino acid mutations (not fully conserved in all seven constructs) per 100 consecutive amino acids, measured as a moving average inside a 100aa window stepping along the sequence. The locations of DSMs are highlighted. The dotted black line shows the impact of omitting one isolate M920 having the $ID1_{DSM}$. **c** A schematic representation of the construct used for protein expression and the workflow for protein production. The extracellular domain of VAR2CSA (NTS-DBL6 or -DLB7, see Table 1 for exact boundaries) was cloned into a mammalian expression vector flanked by an N-terminal signal sequence (red) that directs the protein for export and a C-terminal his tag (yellow) for detection and purification.

**Table 1 Geographic origin, extracellular region architecture, and construct boundaries of VAR2CSA proteins under study.**

| Full-length VAR2CSA allele | Geographic origin | DBL domains | DSM in DBL2[20] | DSM in ID1[21] | Boundary of expressed protein (AA) | GenBank accession number |
|---|---|---|---|---|---|---|
| FCR3 | Gambia | 6 | FCR3-like | 3D7-like | 1-2649 | AAQ73926 |
| NF54 | Netherlands | 6 | 3D7-like | 3D7-like | 1-2642 | XM001350379 |
| HB3 | Honduras | 6 | FCR3-like | 3D7-like | 1-2695 | KOB63403 |
| 7G8 | Brazil | 6 | FCR3-like | 3D7-like | 1-2680 | EUR69480 |
| M. Camp | Malaysia | 6 | 3D7-like | 3D7-like | 1-2673 | OK346630 |
| M920 | Tanzania | 6 | FCR3-like | WR80-like | 1-2670 | OK346631 |
| M200101 | Mali | 7 | FCR3-like | 3D7-like | 1-2963 | QGY73032 |

*DBL Duffy binding-like, DSM dimorphic sequence motif, ID1 inter-domain 1.*

plasmids containing full-length VAR2CSA-FCR3, NF54, HB3, 7G8, M. Camp, M920 or M200101. After 7 days of expression, culture media was loaded on a nickel NTA column. The eluted fractions were concentrated and purified further using 100 kDa cutoff spin concentrators. The proteins were then loaded onto a Superose 6 (S6) size exclusion column (SEC). The exclusion limit of the S6 column is 5000 kDa, thus full-length VAR2CSA would enter beyond the void volume.

The purified full-length protein had high purity and migrated mostly as 1 species under SDS-PAGE non-reducing conditions (Fig. 2a). The NTA fractions after diafiltration (with a 100 kDA cutoff membrane) ran as a single peak in the S6 column (Fig. 2b and Supplementary Data 1), indicating the proteins had high

purity and ran as a single species. Upon reduction with dithiothreitol (DTT), protein migration was altered for most alleles. As predicted for a protein with many disulfide bonds, the denatured but non-reduced protein migrated more quickly than the denatured and reduced form of the FCR3, 7G8, and M920 alleles. Unexpectedly, HB3, M. Camp and M200101 alleles displayed the opposite pattern, and upon reduction migrated more quickly in the gel. Overall, the expressed full-length VAR2CSA recombinants formed disulfide bonds and different alleles varied in their migration patterns.

**High yields of full-length VAR2CSA recombinants were obtained in optimized mammalian expression system.** The

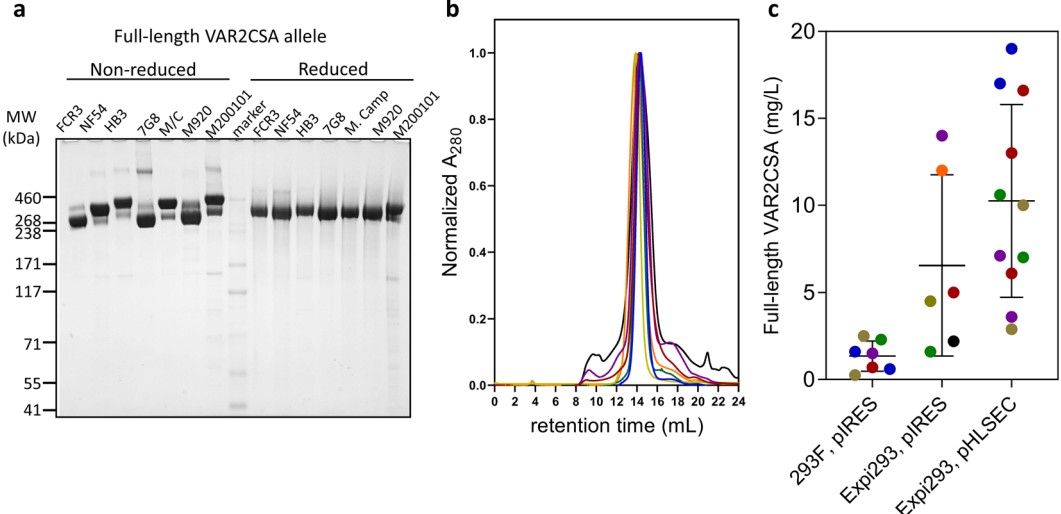

**Fig. 2 Expression and purification of full-length VAR2CSA, including SDS-PAGE analysis and protein yields. a** VAR2CSA recombinants expressed in Expi293 cells were analyzed in SDS-PAGE gels. Ten μg of purified full-length VAR2CSA FCR3 (lane 1 and 9), NF54 (lane 2 and 10), 7G8 (lane 3 and 11), HB3 (lane 4 and 12) M. Camp (lane 5 and 13), M920 (lane 6 and 14), and M200101 (lane 7 and 15) were loaded on a 3–8% tris-acetate gel (with or without 100 mM DTT) and Coomassie-stained. Lane 8 is the molecular weight marker (kDa). The theoretical sizes of full-length VAR2CSA range from 308 kDa to 346 kDa. **b** Size exclusion analysis of all 7 full-length VAR2CSA proteins ran on a Superose 6 column. Each trace is in a different color (FCR3 = blue, NF54 = red, HB3 = purple, 7G8 = yellow, M. Camp = green, M920-orange, M200101 = black). All proteins appear predominantly as a single species that elute around 14 mL (void volume of the column is 8 mL). **c** Different HEK293 suspension cell lines were combined with different expression plasmids. Protein yields for each allele is represented in a different color dot. All the preps ranged from 0.5 to 2L of transfected cells. The average yield of 293F plus pIRES plasmid was 1.4 mg/L ± 0.9, the average yield of Expi293 plus pIRES was 6.6 mg/L ± 5.2, and the average yield of Expi293 plus pHLSEC plasmid was 9.3 mg/L ± 5.9. Error is the standard deviation based on 7, 6, and 11 preps respectively.

initial expression of full-length VAR2CSA was designed according to previously published methods that used suspended HEK293 cells and a plasmid under the control of the CMV promoter[29,30]. Expressing FCR3, NF54, HB3, 7G8, and M. Camp alleles using these methods yielded an average of 1.4 mg/L (ranging from 0.25 mg/L for 7G8 allele to 2.5 mg/L for 7G8 allele). The expression of full-length VAR2CSA proteins was optimized by combining two different approaches: (1) employing Expi293 cells, a new generation of suspended HEK293 cells (ThermoFisher Expi293 manual), and (2) switching the vector to the pHLSEC expression plasmid. The Expi293 expression system is a high-density expression system that supports a 2- to 10-fold increase over traditional HEK293F cells (ThermoFisher Expi293 manual). By combining the Expi293 expression system with the CMV promoter, production of full-length VAR2CSA recombinants for NF54, HB3, 7G8, M. Camp, M920, and M200101 alleles was increased an average of 4.9-fold, ranging from 1.5 mg/L for M. Camp allele to 14 mg/L of the HB3 allele (Fig. 2c, Supplementary Table S1 and Supplementary Data 1). To further increase protein yields of full-length VAR2CSA recombinants, a new promoter was tested. Expression of the pHLSEC plasmid is under control of the β-actin promoter, which is a stronger promoter than CMV and has been shown to support high yields of protein in transient transfection[37]. Together, the use of the pHLSEC plasmid with the Expi293 system increased expression of full-length VAR2CSA recombinants for FCR3, NF54, HB3, 7G8, and M. Camp alleles to an average of 10.3 mg/L yields (ranging from 2.9 mg/L for 7G8 allele to 19 mg/L for FCR3 allele), representing a 7.6-fold increase on average in protein production (Fig. 2c, Supplementary Table S1 and Supplementary Data 1).

**Purified full-length VAR2CSA recombinants are rich in α-helical content and folded.** Protein secondary structure was analyzed by far-UV circular dichroism (CD) spectroscopy for all seven full-length VAR2CSA recombinant proteins (Fig. 3a and

Supplementary Data 1). The far-UV (190-280) spectrum provides information on the secondary structure of proteins in solution by absorbance of polarized light. All seven full-length VAR2CSA proteins displayed α-helical characteristics with spectral minima at 208 and 222 nm. The high α-helical content observed in the far-UV CD spectra is consistent with the full-length structure of VAR2CSA and with the isolated DBL domain structures of VAR2CSA[14,38,39]. When the protein samples were heated to 95 °C the negative ellipticity at 208 and 222 nm disappeared, suggesting denaturation of the secondary structure, and leading to an unfolded conformation (Fig. 3b and Supplementary Data 1).

**Full-length VAR2CSA has high thermal stability that varies between alleles.** Thermal melts were performed to quantify the stability of full-length VAR2CSA and interrogate whether stability varied by allele. The CD signal was monitored as a function of temperature and these values were converted to fraction unfolded (Fig. 3c and Supplementary Data 1), then fitted to determine the melting temperature ($T_m$). $T_m$ is the temperature at which 50% of the protein is unfolded; the higher the $T_m$ the more stable the protein, as it requires greater energy input to unfold. For all seven alleles, as the temperature increased, a sharp transition was observed, indicating unfolding of a correctly folded protein. The $T_m$ of unfolding occurred around 70 °C and this varied by allele (ranging from 70 °C to 75 °C). The NF54 variant was the most stable with a $T_m$ of 75 °C, while M920 was the least stable with a $T_m$ of 70 °C (Fig. 3d and Supplementary Data 1).

**Full-length VAR2CSA binds CSA with high affinity that varies between alleles.** We explored the CSA-binding capacity of the full-length VAR2CSA recombinants expressed in this study by Biacore and direct ELISA. Decorin was immobilized on the sensor chip and increasing amounts of full-length VAR2CSA showed increased binding; from these values the steady-state equilibrium $K_d$ was

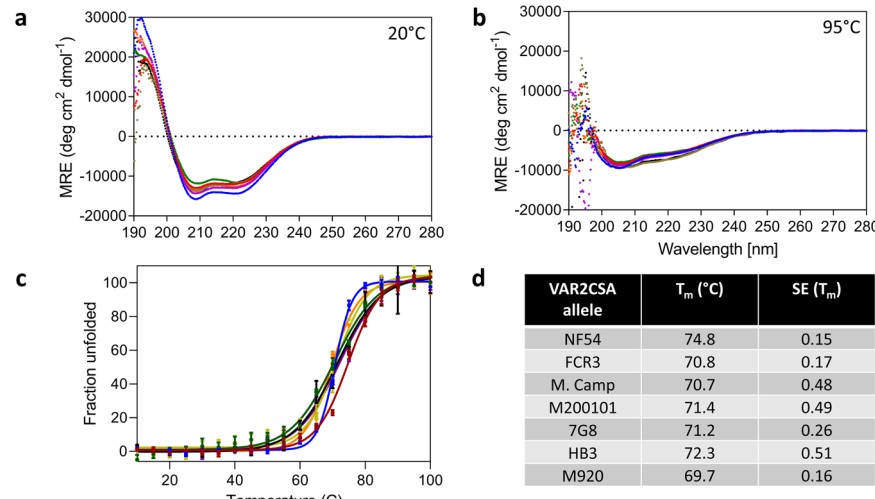

**Fig. 3 Biophysical characterization of full-length VAR2CSA using circular dichroism spectroscopy.** Far-UV CD wavelength scan of 0.2 mg/mL full-length VAR2CSA (FCR3 = blue, NF54 = red, HB3 = purple, 7G8 = yellow, M. Camp = green, M920-orange, M200101 = black) at **a** 20 °C and **b** 95 °C, showing the loss of α-helical structure upon thermal denaturation. **c** Thermal melt of all the seven alleles monitoring the loss of CD signal at 200 nm and plotting the fraction unfolded vs. temperature. **d** The $T_m$ values of all seven alleles obtained by the fit. NF54 is statistically different. Error is SEM based on three biological replicates.

**Table 2 Binding of recombinant full-length VAR2CSA proteins to CSA (decorin).**

| VAR2CSA allele | Glycosylated | | Deglycosylated | |
|---|---|---|---|---|
| | $K_d$ (nM) | SE (KD) | $K_D$ (nM) | SE (KD) |
| 7G8 | 35 | 1.3 | 27 | 0.6 |
| NF54 | 59 | 1.1 | 87 | 0.28 |
| HB3 | 84 | 10.8 | 65 | 0.49 |
| M920 | 70 | 6.1 | 48 | 6.6 |
| M. Camp | 73 | 10.9 | 31 | 5.5 |
| FCR | 860 | 106 | 520 | 53 |
| M200101 | 130 | 0.16 | 61 | 5.4 |

The binding affinity was measured by Biacore for both the glycosylated and deglycosylated full-length VAR2CSA alleles. The $K_D$ was determined from steady-state analysis and the error is SEM based on three replicates.

determined. Since the binding of VAR2CSA to CSA involves multiple DBL domains, a simple 1:1 binding model could not be used and therefore the on and off rates could not be calculated[14,30]. $K_d$ was determined for all seven alleles and the strength of binding varied between different VAR2CSA alleles: full-length VAR2CSA from 7G8 exhibited the strongest binding to CSA with a $K_d$ of 35 nM, while VAR2CSA from M200101 and FCR3 showed the weakest binding with $K_d$ values of 130 and 860, respectively (Table 2, Supplementary Fig. S1 and Supplementary Data 2). In addition, we used direct ELISA to confirm the interaction of VAR2CSA and CSA (Supplementary Fig. S2 and Supplementary Data 1). Previous studies have reported that full-length VAR2CSA binds to CSA with high nanomolar affinity, detected by an ELISA-based assay[30,31] that is unable to detect low-affinity micromolar interactions with the individual VAR2CSA domains[30]. We detected strong binding of our full-length recombinant VAR2CSA in the ELISA format and the same trend was observed between the direct ELISA and Biacore measurements: 7G8 showed the strongest binding while M200101 and FCR3 showed the weakest. The values obtained for NF54 in the previous studies[30] are comparable to the binding affinity presented in this study. This finding indicates that the sequence diversity or structural differences in VAR2CSA variants impacts protein binding affinity to CSA.

**Full-length VAR2CSA recombinants bind specifically to CSA.** To test the binding specificity of full-length VAR2CSA, we measured the binding of these seven alleles to various receptors by ELISA. Decorin, chondroitin sulfate C (CSC), or BSA were coated to the plate, then full-length VAR2CSA (at 5 μg/mL) was detected by an HRP-conjugated anti-his-tag antibody (Supplementary Fig. S3A, S3B and Supplementary Data 1). Binding to decorin was much greater than binding to CSC or BSA, suggesting that all variants were specific to the CSA groups on decorin. Modest binding of some VAR2CSA alleles to the CSC preparation (which was purified from shark cartilage) was observed, possibly due to the high residual CSA fraction in the CSC preparation. The lot of CSC used in these experiments had a ratio of 1.58:1 CSC to CSA. Greater binding to decorin could arise from decorin being adsorbed to the plate at higher levels than CSC. To compare the density of chondroitin sulfate displayed by adsorbed decorin vs. CSC, an ELISA was performed with an anti-chondroitin sulfate antibody that recognizes both CSA and CSC (Supplementary Fig. S3C and Supplementary Data 1). The signal observed is comparable in the decorin and CSC wells, suggesting that the density of chondroitin sulfate is comparable between the two plates and that the greater binding of VAR2CSA to decorin is best explained as selective adhesion to CSA rather than CSC.

**Full-length VAR2CSA recombinants are recognized by naturally acquired antibodies in a gravidity and trimester-dependent manner.** Recognition of recombinant full-length extra-cellular VAR2CSA proteins by naturally acquired antibodies from malaria-exposed individuals was investigated. Using randomly selected plasma samples from 150 Malian pregnant women who were enrolled in a malaria immunoepidemiology cohort (NCT01168271), IgG binding to full-length VAR2CSA recombinants (and MSP1 control) was measured by ELISA. Plasma ELISA IgG reactivity to all full-length VAR2CSA recombinants increased with gravidity (Fig. 4, Supplementary Fig. S4 and Supplementary Data 1) as well as with trimester (Fig. 5, Supplementary Fig. S5 and Supplementary Data 1). Proportions of women who provided samples in first, second, or third trimester, or at delivery, did not differ significantly between gravidity groups ($p = 0.3$, Chi-square test), suggesting a skew in trimester distribution should not affect

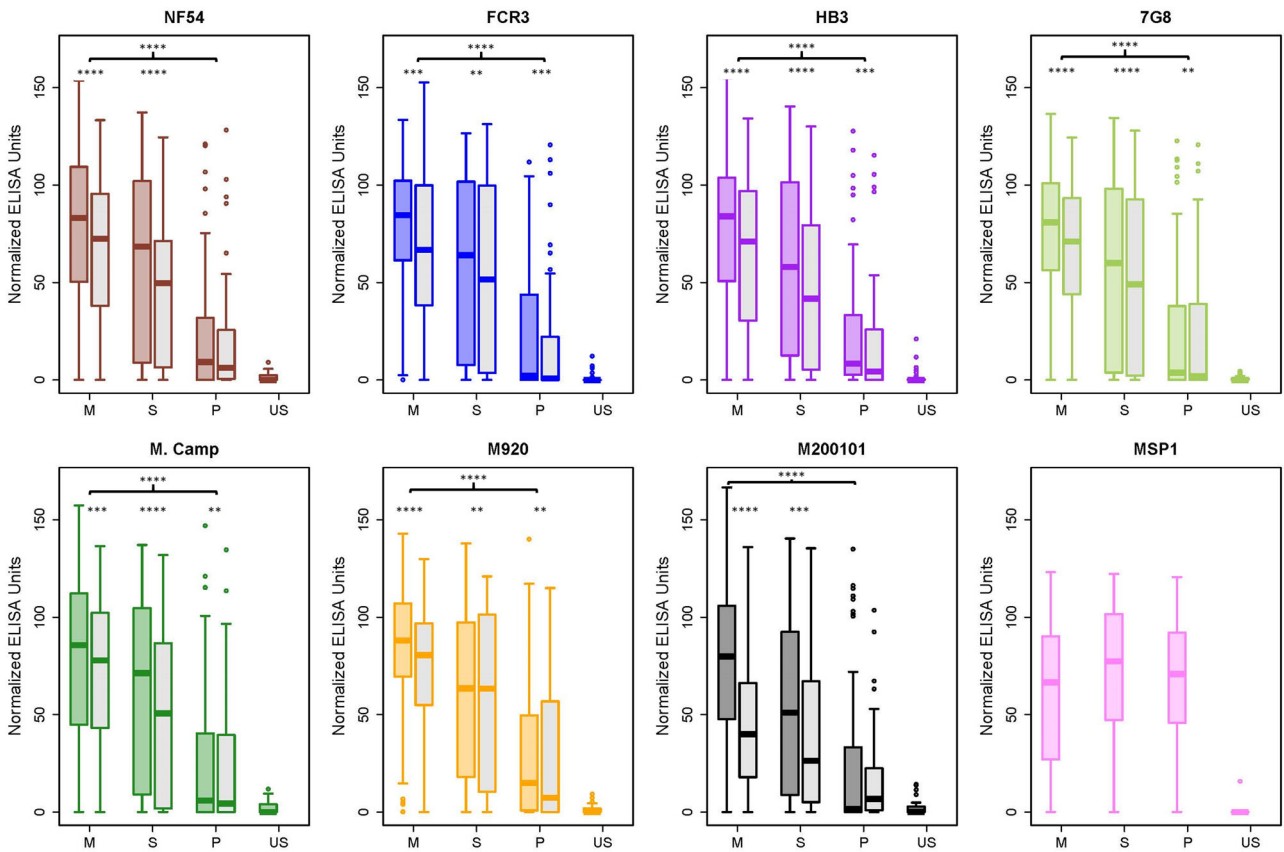

**Fig. 4 Full-length VAR2CSA recombinants are recognized by naturally acquired antibodies in gravidity- and glycosylation-dependent manner.** ELISA plates were coated with full-length VAR2CSA 150 samples from malaria-exposed multigravid (MG), secundigravid (SG), and primigravid (PG) were diluted 1:1000 and measured for reactivity to all full-length VAR2CSA proteins. Box and Whisker representation of ELISA reactivity to VAR2CSA from 150 samples from malaria-exposed women stratified by gravidity ($n = 50$ multigravida, $n = 50$ secundigravida and $n = 50$ primigravida) and 50 US naive samples. The solid line represents the median of reactivity, error bars represent 1.5 times the interquartile range. For each allele (FCR3 = blue, NF54 = red, HB3 = purple, 7G8 = yellow, M. Camp = green, M920-orange, M200101 = black), the glycosylated protein shaded in the corresponding color and the deglycosylated is shaded in gray. The most highly significant difference between MG and PG is displayed with the bracket indicated by **** ($p < 0.0001$); significant differences between MG and PG were identical for both glycosylated and deglycosylated proteins (for a more detailed analysis between the groups see Supplementary Figs. S5 and S6). Statistically significant differences between the values of the glycosylated and deglycosylated are indicated with *$p < 0.05$, **$p < 0.01$, ***$p < 0.001$, and ****$p < 0.0001$. The p-values for both comparisons were generated by the unpaired Mann–Whitney test.

comparisons by gravidity. Using a linear regression model, the relationship of both gravidity and trimester to VAR2CSA antibody levels was highly significant (p-values: $2 \times 10^{-16}$ for gravidity and $2 \times 10^{-8}$ for trimester). However, the magnitude of the effect of gravidity on antibody levels is roughly fourfold higher than trimester ($+23.8$ for gravidity and $+6.8$ for trimester). No gravidity or trimester-related differences in reactivity to MSP1 were observed (Figs. 4, 5 and Supplementary Data 1). Plasma from malaria-naive US individuals showed no reactivity to VAR2CSA and MSP1 recombinant proteins (Fig. 4, Supplementary Fig. S4 and Supplementary Data 1).

**Glycosylation of VAR2CSA has modest effects on affinity towards CSA and recognition by naturally acquired antibodies.** The effects of protein glycosylation on affinity towards CSA and serum antibody recognition were tested. Secreted proteins expressed in mammalian cells like HEK293 are subject to post-translational N-linked and O-linked glycosylation, but proteins expressed in *Plasmodium falciparum* are not. Therefore, these modifications could alter the function or immunogenicity of the recombinant proteins. Recombinant VAR2CSA was enzymatically deglycosylated to preserve the wild-type amino acid sequence and prevent destabilizing effects from introducing many

mutations needed to abrogate all glycosylation sites. Indeed, the glycosylation modifications were removed from the proteins as indicated by SEC (Supplementary Fig. S6A and Supplementary Data 1). The retention volume of the deglycosylated proteins elutes around 15 mL vs. that of glycosylated proteins that elutes around 14 mL. The larger elution volume indicates a smaller size. The decrease in size was also confirmed by SDS-PAGE showing an increased migration of the VAR2CSA protein under both reduced and non-reduced conditions (Supplementary Fig. S6B). Interestingly, similar migration patterns after reduction were observed for the deglycosylated proteins as seen for the glycosylated proteins: the denatured but non-reduced protein migrated more quickly than the denatured and reduced form of the FCR3, 7G8, and M920 alleles, while HB3, M. Camp, and M200101 alleles displayed the opposite pattern, and upon reduction migrated more quickly in the gel. These results suggest that glycosylation is not the cause of the unexpected migration patterns and instead that these are inherent to the sequence and structure of those alleles. Affinity measurement by Biacore demonstrated a slight decrease in $K_d$ of the deglycosylated protein for 6 of the 7 alleles, suggesting enhanced binding to CSA (Table 2 and Supplementary Data 2). In contrast, the NF54 allele showed a slight increase in $K_d$ suggesting reduced binding. Taken together, protein glycosylation could modestly modulate affinity to CSA, however,

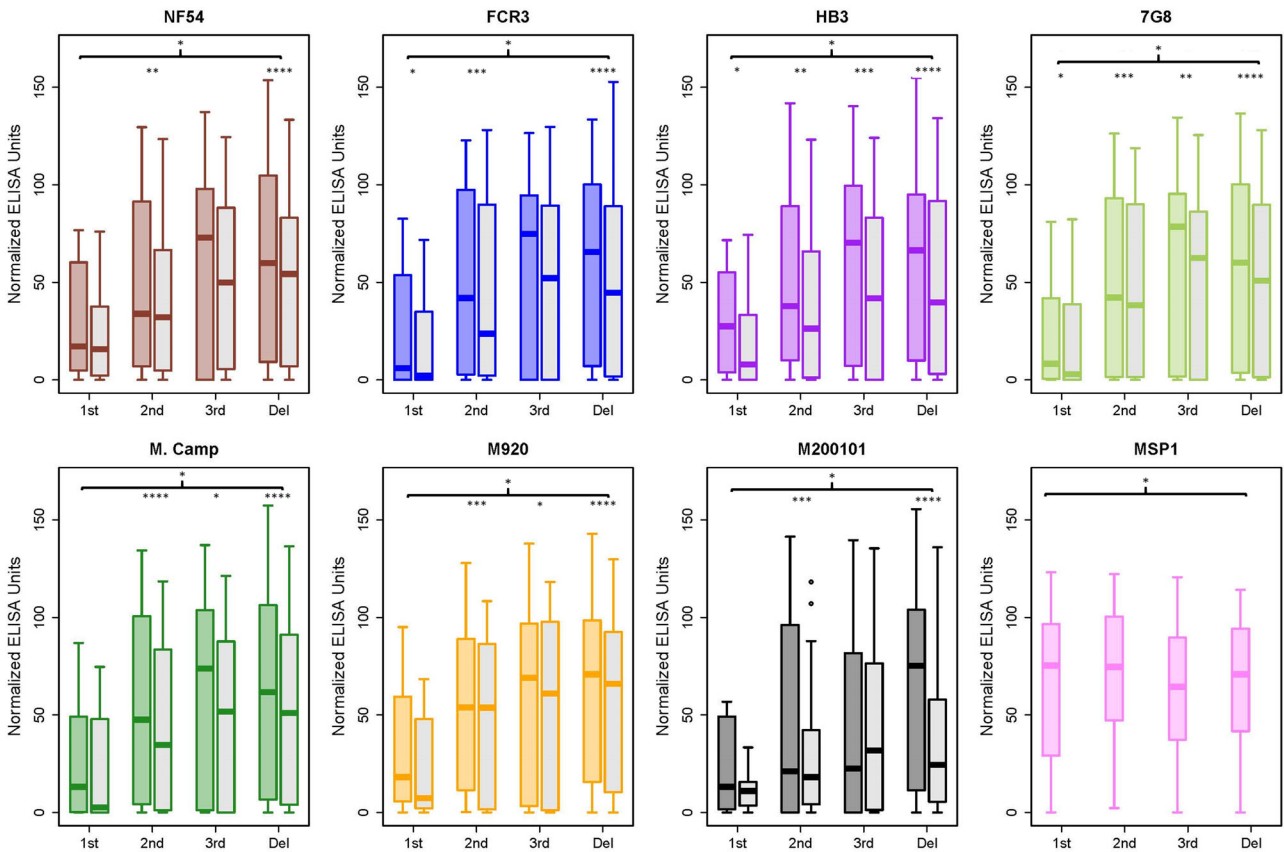

**Fig. 5 Recognition of full-length VAR2CSA recombinants by naturally acquired antibodies increases with trimester.** The same data presented in Fig. 4 were separated into different bins of trimester instead of gravidity. The trimester bins were defined as first (≤13 weeks gestation), second (14–26 weeks), third (≥27 weeks), and time of delivery. Box and Whisker representation of ELISA reactivity to VAR2CSA from 148 samples stratified by trimester ($n = 11$ first, $n = 46$ second, $n = 20$ third, $n = 71$ delivery) from malaria-exposed women and 50 US naive samples. The solid line represents the median of reactivity, error bars represent 1.5 times the interquartile range. For each allele (FCR3 = blue, NF54 = red, HB3 = purple, 7G8 = yellow, M. Camp=green, M920-orange, M200101 = black), the glycosylated protein shaded in the corresponding color and the deglycosylated is shaded in gray. The significant difference between the first trimester and delivery for the glycosylated protein is displayed with the bracket indicated by * ($p < 0.05$). After deglycosylation, significant differences ($p < 0.05$) between first trimester and delivery remained for each allele except for NF54 (for a more detailed analysis see Supplementary Figs. S7 and S8). Statistically significant differences between the values of the glycosylated and deglycosylated are indicated with *$p < 0.05$, **$p < 0.01$, ***$p < 0.001$, and ****$p < 0.0001$. The $p$-values for both comparisons were generated by the unpaired Mann–Whitney test.

this effect could enhance or diminish binding depending on the allele.

The reactivities of glycosylated and deglycosylated full-length VAR2CSA proteins to naturally acquired antibodies were compared, using the same plasma from the 150 Malian pregnant women to measure IgG binding to full-length VAR2CSA recombinants (and MSP1 control) by ELISA. All seven deglycosylated full-length VAR2CSA recombinants displayed the same gravidity-dependent reactivity as the glycosylated variants (Fig. 4, Supplementary Fig. S7 and Supplementary Data 1). Surprisingly, the deglycosylated form of the protein displayed modestly lower recognition by naturally acquired antibodies for all gravidities tested (Fig. 4 and Supplementary Data 1). Reactivity against the deglycosylated protein was also stratified by trimester. Like the glycosylated protein antibody levels against all VAR2CSA alleles increased with trimester (Fig. 5, Supplementary Fig. S8 and Supplementary Data 1). Higher reactivity to the glycosylated rather than the deglycosylated protein was observed across trimesters (Fig. 5 and Supplementary Data 1).

**Reactivity to VAR2CSA recombinants varies between alleles.** Finally, the reactivity of all 150 samples was compared by

VAR2CSA allele. The same ELISA data set described above was stratified by gravidity and glycosylation state. In general, primigravidae samples displayed a unimodal low level of reactivity, secundigravidae displayed a bimodal low and high reactivity, and multigravidae displayed a unimodal high reactivity (Supplementary Fig. S9). The exception to this general pattern was seen with deglycosylated (but not glycosylated) M200101 reactivity, for which secundigravidae and multigravidae seroreactivity was predominantly low. Kruskal Wallis analysis, which tests differences between multiple groups using unpaired data, confirmed significant differences between deglycosylated variants ($p = 0.006$).

Given the paired nature of the data (each serum sample is tested on all alleles), we compared reactivities between alleles by Wilcoxon signed-rank test, revealing numerous differences between alleles that were more frequent and of greater significance with the deglycosylated proteins (exact $p$-values in Supplementary Fig. S10). The pattern of differences seen with primigravidae sera indicated that reactivity was significantly higher to the M920 variant than other variants, using either glycosylated or deglycosylated protein. Conversely, patterns with secundigravidae and multigravidae sera differed dramatically between glycosylated proteins (where differences were few and modest) and deglycosylated proteins (where differences were

frequent and strong). The most consistent pattern was for M200101 reactivity to be lower than all other deglycosylated protein reactivities, using both secundigravidae and multigravidae sera. In addition, the M920 reactivity was significantly higher than several other deglycosylated protein reactivities, particularly with secundigravidae sera. These differences were nearly absent with reactivities measured against glycosylated proteins. Overall, the data indicate that some sequence and structure variation between VAR2CSA alleles could play a role in recognition by naturally acquired antibodies; these patterns differ between gravidity groups with little protective immunity (primigravidae) versus high protective immunity (multigravidae) and can be masked by protein glycosylation.

## Discussion

VAR2CSA is central to the pathogenesis of PM and the acquisition of protective immunity but has been challenging to produce in recombinant form due to its large size, sequence polymorphism, number of cysteine residues, and multidomain architecture. Progress in exploiting VAR2CSA as a target for PM vaccines[23] has been hindered by our poor understanding of immunological and biophysical differences between allelic variants. Here, we address this gap by comparing structural, functional, and antigenic characteristics of seven recombinant full-length VAR2CSA extracellular domains that differ widely in their sequence, geographic origin, and domain architecture. Biophysical characterization of the seven different full-length VAR2CSA proteins produced in this work revealed expected similarities as well as key differences that exist between *var2csa* alleles.

Analysis of recombinant VAR2CSA by CD spectroscopy indicated that all the recombinant full-length VAR2CSA proteins folded properly, have very similar α-helical secondary structures with high secondary structure thermal stability. However, the SDS-PAGE analysis highlighted that alleles of VAR2CSA recombinants differed in migratory patterns in non-reduced or reduced conditions, suggesting differing tertiary structures between alleles. While VAR2CSA alleles give rise to similar folds despite sequence variation, such differences in structure could impact DBL domain-packing or epitope display, influencing VAR2CSA function. Data presented in this study do not rule out that these atypical migration patterns could arise from incorrect folding or disulfide bond formation of the recombinant protein, however, we believe this is unlikely since all three alleles display strong binding to CSA suggesting native function. From a biophysical standpoint, the high thermal stability of all VAR2CSA alleles may prevent the unfolding of secondary structure with loss of conformational epitopes[40] and therefore have benefits for vaccine products. While preserving conformational B-cell epitopes, a hyper-stable protein could prevent unfolding of the antigen in the acidic endosome of antigen-presenting cells, thereby preventing antigen processing and impeding T-cell help[40]. Further studies on full-length VAR2CSA immunogenicity and stability are necessary to determine the optimal stability for the best immune response.

Another striking observation is that the 7 VAR2CSA recombinants bound to CSA with different affinities based on ELISA and SPR analyses, suggesting that some sequence or structural variation may modulate the interaction between VAR2CSA and CSA. The SDS-PAGE migration patterns of 7G8 and FCR3 are similar suggesting these two proteins share structural characteristics, however, 7G8 binds CSA the tightest and FCR3 the weakest out of the seven alleles produced, suggesting some sequence variation modulates binding to CSA. Alternatively, the difference in CSA binding affinity between 7G8 and FCR3 alleles of VAR2CSA could be attributed to tertiary structural differences

that were not observed in our gel migration assay, hence data on high-resolution structures comparing different VAR2CSA alleles are needed to appreciate the impact of structural differences on CSA-binding capability of VAR2CSA with regard to PM pathology. In addition, understanding the differential binding affinity of various alleles of VAR2CSA to CSA expressed on cancer cells may also be relevant to cancer research, where the use of VAR2CSA as a platform for cancer diagnostics and therapeutics is under development[41,42]. A VAR2CSA allele that binds to CSA more tightly could improve targeting of VAR2CSA to CSA on cancer cells.

All full-length proteins generated here were recognized by serum antibodies from Malian pregnant women in a gravidity-dependent manner. Intriguingly, reactivity to different alleles of VAR2CSA varied, with no apparent relationship between levels of serum recognition and CSA-binding strength or protein size. Reactivity differences between alleles could arise from sequence and structural variation that impacts epitopes or their accessibility. One notable sequence variation in this study is an extended VAR2CSA variant that contains a 7th DBL domain[34]. Interestingly, antibodies from both multigravid and primigravid women reacted to this atypical VAR2CSA at lower levels than the other alleles. The extra DBL domain could further influence the structure and hence epitope display, perhaps by restricting access to some conserved epitopes in the M200101 variant. A second notable sequence variation is the VAR2CSA allele M920, which has a dimorphic sequence in ID1 which contributes to much of the sequence variation amongst the seven different alleles (Fig. 1b). In contrast to M200101, the M920 recombinant protein had higher levels of reactivity to serum, suggesting the alternative dimorphic region did not hinder antibody recognition in the same manner as an extra DBL domain.

*P. falciparum* proteins are not N-glycosylated but many higher-ordered expression systems will add this post-translation modification. Thus, these non-native N-linked glycans could impair protein function or block critical epitopes. Therefore, both glycosylated and deglycosylated proteins were systematically investigated for their binding to CSA and recognition by naturally acquired antibodies. Our results showed minor differences in such effects. Whereas we expected deglycosylation might increase recognition by exposing epitopes on the native protein, we instead observed that deglycosylation decreased recognition of all VAR2CSA alleles by naturally acquired antibodies and this effect was seen across all gravidities. One possible explanation is that glycan modifications decrease the fluctuation of the local protein structure[43,44], which in turn preserves the structure of epitopes and enhances antibody binding.

These results are important since there is no consensus in the literature regarding which glycosylation state of protein to use. In a previous report, full-length VAR2CSA from NF54 allele was generated with all N-glycosylated sites mutated to block glycosylation[30] while the FCR3 variant was generated with all sites intact[28,31]. Both approaches generated full-length VAR2CSA proteins that bound CSA without investigating the impact of N-linked glycans on VAR2CSA function. The results presented here suggest that either approach is valid for generating recombinant full-length VAR2CSA. Both glycosylation states of the protein are functional and have minor differences in binding to CSA. While naturally acquired antibodies have a reduced binding to the deglycosylated form, this reduction is observed across all gravidities of pregnancy, suggesting the affected epitopes are recognized by all women. The M200101 variant offered a striking exception to this general pattern, where deglycosylation markedly reduced seroreactivity only for secundigravidae or multigravidae, suggesting a specific impact on epitopes for protective antibody responses.

If the 7th DBL domain does shield epitopes, then it could play a role in immune evasion allowing the parasite to escape protective immunity. Recent work on the structure of VAR2CSA demonstrated that NTS through ID3, including DBL1-4, forms a core structure with 2 CSA binding channels demonstrating the importance of DBL1-4 in the adhesion of the parasite to CSA[14]. DBL5-6 comprises a flexible arm that flanks the stable core. Speculatively, an additional DBL7 domain that extends the flexible arm could confer steric hindrance of protective epitopes normally exposed with a typical VAR2CSA. Structural details of atypical full-length VAR2CSA variants could elucidate this mechanism and inform vaccinology targeting these atypical VAR2CSA proteins.

Current VAR2CSA-based vaccine design relies on identifying functional fragments of VAR2CSA that bind CSA with high affinity and induce broadly neutralizing antibodies[23]. Two VAR2CSA-based vaccine candidates (PAMVAC and PRIMVAC) have been shown to be safe and to induce functional activity in malaria-naive and malaria-exposed women, but with limited strain-transcending activity[26,27]. We predict that full-length VAR2CSA displays epitopes targeted by strain-transcending antibodies, but the recombinant immunogen may not induce such antibodies. Therefore, we are now positioned to use full-length VAR2CA to purify strain-transcending activity mediated by multigravid antibodies. The highly diverse sequence and domain architecture of VAR2CSA alleles generate variant-specific immune responses that may delay or impair the acquisition of protective variant-transcending antibodies and might also contribute to the variable outcomes of pregnant women with *P. falciparum* malaria.

The sequence diversity in VAR2CSA confounds the ability to develop a VAR2CSA-based vaccine capable of inducing broadly neutralizing antibodies. In previous reports[32,45], 3D7 and FCR3 full-length VAR2CSA proteins were unable to induce broadly neutralizing antibodies in rats or rabbits. These observations could suggest that a single variant of full-length VAR2CSA protein may not be sufficient to generate broadly neutralizing activity of antibodies such as those that develop in PM-resistant multigravidae[32]. Therefore, protective immunity elicited by different full-length VAR2CSA proteins will need to be assessed in future investigations. The full-length VAR2CSA recombinant proteins from seven diverse alleles are now under preclinical exploration to determine whether broadly neutralizing antibodies may be generated by combining full-length VAR2CSA immunogens from multiple alleles.

Further rationale for generating full-length VAR2CSA proteins is that strain-transcending antibodies could recognize epitopes that are only present in the full-length protein. In fact, a variant-transcending human monoclonal antibody (PAM1.4) derived from a B cell of a malaria-immune African woman indicates that some VAR2CSA epitopes are generated only by full-length antigen[46]. Importantly, while PAM1.4 recognizes an epitope displayed by full-length VAR2CSA, PAM1.4 does not exhibit anti-adhesion activity. In addition, we previously reported that individual domains of VAR2CSA from different alleles could not deplete broadly neutralizing activity contained in the plasma of multigravid women[47]. If full-length VAR2CSA specific epitopes responsible for broadly neutralizing activity are displayed by different alleles, then some combination of full-length VAR2CSA recombinants should purify functional activity seen in antibodies from multigravid women.

Finally, a particularly challenging aspect of full-length VAR2CSA expression has been its low yield. Here, a sevenfold increase in expression levels was achieved by using a stronger promoter and a next-generation HEK293 cell line. Our reported yields did vary notably, suggesting additional factors such as sequence variation may play a role in expression levels and affect yield. Our data indicate FCR3 allele achieves the highest yields while HB3 and M200101 achieve the lowest yields. The addition of an extra DBL domain (~300 amino acids) to a large protein could also reduce expression. However, the other six full-length sequences have similar sizes, thus nucleotide or amino acid composition must affect expression levels despite the fact that all sequences were codon-optimized. Furthermore, it appears that sequence variation is not the only factor in expression yields, as 2 runs of NF54 expression resulted in 6 and 17 mg/L yields. These inconsistencies may arise from passage number and cell count during passage and transfection. In general, early cell passages result in higher yields as seen in the above NF54 case. We hypothesize that strict control of passage number and cell count can give consistent high levels of full-length VAR2CSA expression. If the full-length VAR2CSA is to be considered a vaccine construct for a PM vaccine, identifying sequences that maximize protein yield could be critical to generate enough material.

In conclusion, this report provides the first description of the biophysical and immunological differences of seven alleles of full-length VAR2CSA recombinant proteins. Our study demonstrates that the seven alleles have different structural characteristics, bind to CSA with different affinities, and are differentially recognized by antibodies from both primigravid and multigravid malaria-exposed women. Future studies that relate biophysical and immunological characteristics of VAR2CSA variants to clinical outcomes in pregnancy malaria would inform the design of improved VAR2CSA vaccines.

## Methods

**Collection of Plasma samples**. Plasma samples were obtained from participants in the immuno-epidemiology (IMEP) study conducted in Ouéléssébougou (Mali). A detailed description of the IMEP study has been previously reported[48]. Briefly, pregnant women were enrolled between November 2010 and October 2013 into a longitudinal cohort study of mother-infant pairs . The study site is located 80 km south of Bamako, an area of intense seasonal malaria transmission during the rainy season from July to December. Pregnant women aged 15–45 years without clinical evidence of chronic or debilitating illness were asked to participate in the study and gave signed informed consent after receiving a study explanation form and oral explanation from a study clinician in their native language. The protocol and study procedures were approved by the Institutional Review Board of the National Institute of Allergy and Infectious Diseases at the US National Institutes of Health (ClinicalTrials.gov ID NCT01168271), and the Ethics Committee of the Faculty of Medicine, Pharmacy and Dentistry at the University of Bamako, Mali.

For this work, 150 plasma samples collected from Malian pregnant women in different trimesters or at delivery were randomly selected for the screening of antibody reactivity to the different full-length VAR2CSA. Samples from 36 US malaria-naive individuals were also included as controls in the serology assays.

**Construction of full-length VAR2CSA sequences and phylogenetic analysis**. Full-length cDNA sequences of exon 1 were generated from whole-genome sequencing (WGS) Illumina reads using the Consensus Protein Pileup (CPP) tool in R package 'DuffyNGS'[34]. Sequence similarity was measured as edit distance which was determined as the number of mismatched residues of amino acid, using R function 'adist()', Phylogenetic trees were generated using function 'plot.phylo()' from the R package 'ape'. Multiple sequence alignments (MSA) were generated using MAFFT version 7.245[49]. Prevalence of sequence variability was measured by counting the number of non-fully conserved amino acids per 100 adjacent amino acids in the MSA result, repeatedly shifting the 100aa window along the full length of exon 1.

**Plasmid construction**. Full-length VAR2CSA sequences from seven alleles were synthesized with mammalian optimized codons by Genscript (FCR3, NF54, HB3, 7G8, M. Camp, and M920 alleles) and Genewiz (M200101 allele). The boundaries of all the full-length VAR2CSA constructs include NTS to DBL6 or DBL7 (for M200101) domain (Table 1). The codon-optimized sequences were cloned into pIRES2-AcGFP with primers (Sigma Aldrich) containing *NheI* and *BamHI* overhangs by a PCR reaction (Q5, NEB) to amplify VAR2CSA. The PCR product was cloned into the *NheI* and *BamHI* sites of pIRES2-AcGFP (Clonetech) using restriction digestion and ligation with T4 DNA ligase (NEB). Cloning into pHLSEC plasmid construction was similarly performed with modification to the primers that contained *AgeI* and *KpnI* sites . The cloned plasmid DNA was transformed into NEB Stable Competent *E. coli*, DNA was prepared from single colonies and screened for the correct insert by restriction digestion and sequencing. Endotoxin-free DNA of the cloned plasmids were prepared

for large scale production of full-length VAR2CSA vectors using an endotoxin-free high-speed giga prep kit (Qiagen).

**Protein expression in HEK293F**. Typically, HEK293F (Thermo Fisher) suspension cells in 293F Expression media (Thermo Fisher) were grown at 37 °C and 8% $CO_2$, maintaining cultures at continuous log phase growth ($1.0–3 \times 10^6$) for 3–4 passages after thawing. For transfection, 500 mL of culture was seeded at a density of $1.5–2 \times 10^6$ cells/mL in a 2 L flask. For every 500 mL culture transfected, 0.5 mL of 293Fectin (Thermo Fisher) and 0.5 mg of plasmid DNA were used. 500 μL of 293Fectin was diluted slowly into 12.5 mL 293F Expression media, gently mixed and incubated at room temperature (RT) for 5 min. The DNA was diluted into 12.5 mL 293F Expression media and filter sterilized through a 0.2 micro filter. The 293Fectin mixture was slowly added to the DNA mixture, gently mixed and incubated at RT for 15 min. The 293Fectin and DNA mixture was added slowly with shaking to the culture, incubated at RT for 15 min and returned to the incubator at 37 °C and 8% $CO_2$.

**Protein expression in Expi293**. Typically, Expi293 (Thermo Fisher) suspension cells in 293 F Expi293 Expression medium (Thermo Fisher) were grown at 37 °C and 8% $CO_2$, maintaining cultures at continuous log phase growth ($3.0–5 \times 10^6$) for 3–4 passages after thawing. The day before transfection, 500 mL of culture was seeded at a density of $2.5–3 \times 10^6$ cells/mL in a 2 L flask. The day of transfection, cells were diluted back to $2.5–3 \times 10^6$ prior to transfection. Expi293 cells were transfected using 1.4 mL of ExpiFectamine (Thermo Fisher) and 0.5 mg of plasmid DNA per 0.5 L of cells. Plasmid DNA was diluted into 25 mL of OptiMEM (Thermo Fisher) and filter sterilized through a 0.2 micro filter. The ExpiFectamine was slowly added to 25 mL of OptiMEM, gently mixed, and incubated for 5 min. The diluted ExpiFectamine was added slowly to the diluted DNA, gently mixed, and incubated at RT for 10–20 min. The mixture was added to the cells slowly while swirling the flask. The flask was returned to the incubator at 37 °C and 8% $CO_2$ for 16–20 h. The following day, both enhancer I and II (Thermo Fisher) were added to the Expi293 cultures and returned to the incubator.

**Protein purification**. Cultures at 7 days post-transfection were centrifuged at $10,000 \times g$ for 30 min. The spent culture media was sequentially filtered through 0.45 and 0.2 μm filters. The clarified spent media was loaded on a 5 mL HisTrap Excel NTA column (GE Life Sciences). The column was washed with 20 column volumes of wash buffer (20 mM sodium phosphate, 0.5 M NaCl, 0–30 mM imidazole, pH 7.4). The bound protein was eluted with a step gradient of elution buffer (20 mM sodium phosphate, 0.5 M NaCl, 500 mM imidazole, pH 7.4). The eluted NTA fractions were confirmed to contain full-length VAR2CSA proteins by SDS-PAGE (3–8% Tris-acetate, Thermo Fisher) and Coomassie staining. The fractions were pooled and concentrated with a 100 kDa cutoff centrifugal filter unit (Millipore Sigma). The concentrated NTA fractions were dialyzed three times against PBS. The concentrated NTA fractions were loaded onto a Sephrose 6 SEC (Cytivia). Fractions were pooled and concentrated. The absorbance at 280 nm was determined and the concentration of the recombinant VAR2CSA was calculated from the predicted extinction coefficient from the protein sequence. Proteins were then aliquoted and stored at −80 °C.

**Production of degylcosylated VAR2CSA**. Purified VAR2CSA was treated with Protein Deglycosylation Mix II (New England Biolabs), a combination of four enzymes that remove all N-linked and O-linked glycosylation modifications. After 3 h at room temperature, the mix was purified by NTA resin and then S6 SEC. Fractions were pooled and stored at −80 °C until use.

**Circular dichroism spectroscopy**. All expressed full-length proteins were diluted into ultrapure water to a final concentration of 0.2 mg/mL. A Jasco J-815 spectropolarimeter was used to collect far-UV spectrum in a 1 mm quartz cuvette from 190 to 280 nm with 1 nm steps, 2 s average time, and a 1 nm bandwidth. Experiments were performed at both 20 °C and 95 °C. Buffer blanks were collected and subtracted out. Mean residue ellipticity (MRE) was calculated using the following equation: $MRE = (100*\Theta)/(C*n*l)$, where $\Theta$ is the buffer subtracted ellipticity, C is the concentration in mM, n is the number of residues and l is the cuvette path length. Protein thermal melts were measured by collecting the same wavelength scans at temperatures from 10 to 100 °C in intervals of 5 °C. Ellipticity at 200 nm was converted to fraction folded by the equation $Fu = (\Theta_T-\Theta_F)/(\Theta_U-\Theta_F)$, where $\Theta_T$ is the ellipticity at the temperature, $\Theta_F$ is the ellipticity of the folded state and $\Theta_U$ is the ellipticity of the unfolded state. $\Theta_F$ and $\Theta_U$ were calculated by averaging a few points before the transition for $\Theta_F$ and a few after for $\Theta_U$. The fraction unfolded was then plotted against the temperature and fit to the Boltzmann-sigmoidal function to obtain the $T_m$.

**Affinity of full-length VAR2CSA recombinants binding to CSA by surface plasmon resonance (SPR)**. The binding affinity of the VAR2CSA recombinant proteins to CSPG was measured using a Biacore T200 system (GE Healthcare). Briefly, decorin (Sigma) was biotinylated non-specifically at lysine residues in the protein core using Sulfo-NHS-LC-Biotinylation Kit (Thermo). NeutrAvidin protein (Thermo Scientific) was covalently immobilized on a Series S sensor chip CM5 (GE Healthcare) using N-hydroxysuccinimide (NHS) amine coupling chemistry, in flow cell 4 (Fc-4), while a biotin-labeled Pfs25 recombinant was similarly immobilized in the reference Fc-3. Protein binding affinity to the immobilized CSPG was obtained by subtracting the response on reference Fc-3 from that on Fc-4, after injection of a serial dilution of the full-length VAR2CSA recombinants. The affinity analysis was performed in Biacore T200 evaluation software (version 3.2) and the steady-state model was used to calculate the dissociation constant $K_D$.

**Binding of full-length VAR2CSA to glycosaminoglycans by ELISA**. The specificity of full-length VAR2CSA recombinants binding to CSA displayed on decorin (Sigma, D8428) as compared to CSC (Sigma, C4384) and BSA was assessed by ELISA as previously described[34]. Hundred microliters of decorin at 5 μg/mL or 50 μg/mL of CSC and BSA were coated on 96-well plates and incubated overnight at 4 °C. The antigen-coated plates were blocked with PBS 1% BSA for 2 h at 37 °C with shaking. His-tagged VAR2CSA recombinants at 5 μg/mL and MSP1 (used as a non-CSA binding control antigen) were added to triplicate wells and incubated for 2 h at 37 °C with shaking. Plates were washed and 100 μL of anti-His HRP-conjugated antibody (at 1:3000 dilution) were added to each well followed by 1-h incubation at 37 °C with shaking. After washing the plates, 100 μL of 3,3′,5,5′-tetramethylbenzidine (TMB) (Thermo Fisher) was added and the reaction was stopped with equal volume of stop solution (SeraCare) after 10 min. OD was read at 450 nm using the MultiskanFC (Thermo Fisher) plate reader. The surface density of chondroitin sulfate on adsorbed decorin and CSC was tested using the same protocol above, however, the chondroitin sulfate was detected with anti-chondroitin sulfate (Abcam, ab11570) at a concentration of 2 μg/mL and detected with anti-human IgM secondary conjugated to HPR (Abcam, ab97230) at a dilution of 1:4000.

**ELISA reactivity of antibodies to full-length VAR2CSA**. 150 plasma samples collected from malaria-exposed pregnant women (multigravidae, secundigravidae, primigravidae) in Mali were randomly selected for assay and compared to 36 samples collected from US malaria-naive individuals. Reactivity of plasma IgG to the different full-length VAR2CSA was assessed by ELISA as previously described[34]. VAR2CSA recombinants were coated at 1 μg/mL on 96-well plates and incubated overnight at 4 °C. Plates were blocked with buffer containing 0.5 M sodium chloride, 1% TritonX, and 1% bovine serum albumin in PBS and incubated at RT for 2 h, before adding plasma samples diluted at $5 \times 10^{-2}$ in duplicated wells for 1-h incubation at RT. After washing the plates, HRP-conjugated anti-human IgG antibody (at 1:3000 dilution) was added for 1-hr incubation at RT, and a final plate wash was performed. TMB (SeraCare) was added to the wells followed by equal volume of stop solution (SeraCare) after 10 min of incubation at RT in dark. OD was read at 450 nm using the MultiskanFC (Thermo Fisher) plate reader. A pooled hyperimmune plasma from samples of multigravida Malian women (positive control) and a pool of US naive sera (negative control) were included in each plate. Each sample was run in two replicate wells, then their values averaged by arithmetic mean. Normalized ELISA values were obtained by using the equation $100 \times [(OD_{sample} − OD_{US\ naive\ pool})/(OD_{Multigravid\ pool} − OD_{US\ naive\ pool})]$.

**Statistics and reproducibility**. All statistical assessments of ELISA antibody reactivity were done using R version 3.6.3. Comparison tests between gravidities and between alleles used Wilcoxon rank sum tests, with Benjamini & Hochberg correction for multiple comparisons.

**Reporting summary**. Further information on research design is available in the Nature Research Reporting Summary linked to this article.

## Data availability
The amino acid and DNA sequences of VAR2CSA from isolates M. Camp and M920 used in this article, as well as the full-length amino acids sequences of VAR2CSA from isolates FCR3, NF54, HB3, 7G8, and M200101 are accessible from their GenBank accession numbers provided in Table 1. All data relevant to this study are available from the corresponding author on reasonable request.

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

## Acknowledgements
The authors thank J. Patrick Gorres for assistance in preparing and editing this manuscript, John Andersen for helpful discussion of Biacore experimental setup and data analysis, Puthupparampil Scaria, David Narum, Karine Reiter for helpful discussions. This work was supported by the Intramural Research Program of the National Institute of Allergy and Infectious Diseases, National Institutes of Health.

## Author contributions
J.P.R., J.D. and P.E.D. designed the study. J.P.R., J.D., B.C.M.T., M.B. and M.V.C. performed the experiments. R.D.M. performed bioinformatic analysis. M.F. and P.E.D. collected the plasma samples. J.P.R., J.D., R.D.M. and P.E.D. analyzed the data. J.P.R, J.D. and P.E.D wrote the manuscript. R.M. and N.H.T. provided the pHLSEC plasmid and experimental guidance to increase expression and purity. All authors participated in the discussion and manuscript editing.

## Competing interests
The authors declare no competing interests as defined by Nature Research, or other interests that might be perceived to influence the results and/or discussion reported in this paper.
