## [Transparent Peer Review File · Communications Biology]

Reviewers' comments:

Reviewer #1 (Remarks to the Author):

The manuscript "Allelic variants of full-length VAR2CSA, the placental malaria vaccine candidate, differ in antigenicity and receptor binding affinity" by Dr Renn and colleagues is an interesting study that will be of interest for the community.

Indeed, in this manuscript the authors have expressed and characterized 7 different full length VAR2CSA variant in the HEK293 cell expression system.

The manuscript is well written; however, I have some concerns on some of the data that can be easily answered.

Figure 2A. The authors have loaded an SDS PAGE gel with reduced and non-reduced purified VAR2CSA recombinant proteins. As mentioned by the authors, the migration pattern for FCR3, 7G8 and M920 is normal (non-reduced protein migration quicker than reduced), this is not the case for the other VAR2CSA variants. As a general rule it is always better to separate reduced from non-reduced samples to avoid reducing agent leakage between wells. Could it be the reason for this strange pattern?

Figure 2B. The yield for the orange dot (M920?) can not be found in the 293F, pIRES and Expi293, pHLSEC expressed proteins. Although the increased level of expression is easy to observe between the 293F pIRES and the Expi93 pHLSEC panels, it is hard to conclude on the improvement obtain using the pHLSEC instead the pIRES plasmid since only two proteins were expressed with the combination Expi293 and pIRES. I would also favor a table instead a figure which will make it easier to compare the production yield for each VAR2CSA variant

Figure 3B. Although the binding properties of the different var2CSA proteins were assessed on the CSA-bearing bovine glycoprotein decorin immobilized on plastic by ELISA, I would recommend to assess the binding specificity of these proteins using also CSA and chondroitin sulphate C (CSC). Indeed, it is well known that parasites expressing VAR2CSA bound to CSA and not CSC.

Finally, the ELISA is not a good method to measure the affinity of a protein to its receptor. Surface plasmon resonance (SPR) experiments should be performed to measure the association strength of the different VAR2CSA to decorin and to provide the affinity constant K_D (K_{off}/K_{on}).

Finally, as mentioned by the authors in the discussion, the expressed recombinant proteins are likely to be glycosylated since they didn't remove the N-glycosylation sites. Could these glycosylations affect the binding to CSA and the sera recognition of the different recombinant proteins? It could be interesting to perform deglycosylation experiments to assess the impact of glycosylation on the VAR2CSA binding properties.

Reviewer #2 (Remarks to the Author):

In the manuscript "Allelic variants of full-length VAR2CSA, the placental malaria vaccine candidate, differ in antigenicity and receptor binding affinity" Renn et. al., describes the production of 7 full length VAR2CSA recombinant proteins. In 2010 Srivastava published structural and biochemical data on the 3D7 full length VAR2CSA and Khunrae on the FV2 genotype. The present study extends the analyses to 5 more genotypes.

In general the study is not very thorough and adds only little to the understanding of the structural features of VAR2CSA and pathology of this type of malaria.

Specific comments:

1) Figure 1: The manuscript presents a phylogenetic tree over a panel of full length VAR2CSA sequences. It is unclear how the tree is build, i.e. what type of tree is it. But more importantly with the block structure variation seen in PfEMP1 sequences it is meaningless to make a phylogenetic tree over the entire sequence. If the authors want to pursue this in a ligand binding context, they could have built a more reliable tree over the ligand binding region in DBL2 rather than the entire sequence.

2) Figure 2: The authors state that they have "high purity protein". A silver stain or HPLC would reveal high level of impurities. Even with the current coomassie stain it is evident that quite a few of the proteins (7G8, MC, M0101) have a significant amount of impurities. A conservative estimate would be that for example 7G8 is only 50% pure, this makes comparison of yields meaningless when level of impurities differ that much. And as no SDS page is provided for Figure 2B it is not possible to evaluate comparative yields. As the proteins are only purified in a one step IMAC procedure, one would expect many impurities, and makes OD measurements of concentrations unreliable. Preferably the proteins should have been purified using a two step procedure (IMAC followed by IEX or SE). More worrying is the fact that not all the proteins change migration pattern after reduction, clearly one would expect a slower migration of these highly disulfide bonded proteins. This questions the quality of the proteins that doesn't follow this pattern. This finding is mentioned by the authors in the manuscript.

3) Figure 3A: A two point measure using CD doesn't make much sense. It would be standard to show a full melting curve to see any differences among the proteins.

4) Figure 3C. I assume that the assay is done with a titration of the protein and a titration of decorin as stated on the legend? One of the main point in the manuscript is that the different VAR2CSA proteins have different binding phenotypes, as described in Fig3C. This is a very superficial analyses, and seemingly several of the proteins have a flat low curve indicating unspecific binding. The assay lacks controls for specificity and to report fine differences I would suggest to use label free kinetic analyses (ITC or Biacore).

5) Figure 5. With limited efforts the authors could have provided a proper serological analyses. Using a single concentration of a pooled sample is meaningless. The authors have previously excelled in serological analyses of PfEMP1 proteins and would have sera ready available. At least end point titrations and subtypes could have done with the pools. And finally I wonder why the authors have not immunized with the proteins and shown the capacity of the immunogens to inhibit CSA binding of different genotypes of VAR2CSA binding parasites.

Reviewer #3 (Remarks to the Author):

The manuscript by Renn et al., contain important information that help us to understand placental malaria and progress in the design of protective vaccines to the vulnerable populations. They show high yield expression and purification of recombinant full length extracellular VAR2CSA antigen which produce 7 alleles. They compare structural, functional and antigenic characteristics and show that they vary in their binding to CSA. The rationale for focusing on the 7 fragments is well stated and the work helps move the malaria field forward.

1. Authors report some inconsistencies in the NF54 yield from 2 different runs (6 and 17 mg/mL), which they hypothesize that may be due to passage numbers and cell count, I think performing this experiment to confirm that it is true would make the manuscript much stronger.

2. It would be better to classify the serum reactivity of the recombinants based on trimester of the pregnancy in the primigravid and multigravid groups and if this should show differences between the two groups as indicated in figure 4.

3. There is no indication of the kind of statistics that were performed for data that show error bars.

4. While the authors state that US controls were used, it is not clear in Fig 4 the data showing this controls or how they were incorporated in the analysis.

Reviewer #1 (Remarks to the Author):

The manuscript "Allelic variants of full-length VAR2CSA, the placental malaria vaccine candidate, differ in antigenicity and receptor binding affinity" by Dr Renn and colleagues is an interesting study that will be of interest for the community. Indeed, in this manuscript the authors have expressed and characterized 7 different full length VAR2CSA variant in the HEK293 cell expression system. The manuscript is well written; however, I have some concerns on some of the data that can be easily answered.

Figure 2A. The authors have loaded an SDS PAGE gel with reduced and non-reduced purified VAR2CSA recombinant proteins. As mentioned by the authors, the migration pattern for FCR3, 7G8 and M920 is normal (non-reduced protein migration quicker than reduced), this is not the case for the other VAR2CSA variants. As a general rule it is always better to separate reduced from non-reduced samples to avoid reducing agent leakage between wells. Could it be the reason for this strange pattern?

Response: We reran the SDS-PAGE as the reviewer suggested, grouping the non-reduced samples and separating them from the reduced samples. With our new analysis, the conclusions remain the same - that some alleles run slower upon reduction and others run faster. The figure was updated in the manuscript (please see **Figure 2A**).

Figure 2B. The yield for the orange dot (M920?) cannot be found in the 293F, pIRES and Expi293, pHLSEC expressed proteins. Although the increased level of expression is easy to observe between the 293F pIRES and the Expi93 pHLSEC panels, it is hard to conclude on the

improvement obtain using the pHLSEC instead the pIRES plasmid since only two proteins were expressed with the combination Expi293 and pIRES. I would also favor a table instead a figure which will make it easier to compare the production yield for each VAR2CSA variant

Response: In the revised manuscript, we provide a table summarizing the expression data (please see **Supplementary Table 1**) in addition to the graph in **Figure 2C**.

We have generated additional expression data from 3 more preparations of VAR2CSA using pIRES+Expi293 (please see **Figure 2C** and **Supplementary Table 1**: HB3, 7G8 and M/C). With these additional data included, we find using pIRES+Expi293 has a 4.9-fold increase of yields over pIRES+293F. When Expi293 was combined with pHLSEC, we observed a 7.6-fold increase over pIRES+293F. We concluded that both improvements increase VAR2CSA yields.

The expression of M920 in the Expi293+pHLSEC was tested, however, the combination of the M920 sequence in the pHLSEC backbone resulted in toxicity to the Expi293 cells. Consequently, this combination could not be evaluated.

Figure 3B. Although the binding properties of the different var2CSA proteins were assessed on the CSA-bearing bovine glycoprotein decorin immobilized on plastic by ELISA, I would recommend to assess the binding specificity of these proteins using also CSA and chondroitin sulphate C (CSC). Indeed, it is well known that parasites expressing VAR2CSA bound to CSA and not CSC.

Response: In the revised manuscript, we include ELISA data showing binding of all 7 VAR2CSA alleles (both glycosylated and deglycosylated forms) on other glycosaminoglycans, including heparin sulfate and chondroitin sulfate C (please see **Supplemental Figure 3** below). As expected, we observed a high level of glycosylated VAR2CSA binding to decorin for all alleles, and weaker binding to CSC with the M200101 allele. A similar high level of binding was observed with deglycosylated VAR2CSA, again with a modest level of background binding for some alleles to heparin sulfate and chondroitin sulfate C. Background binding to the CSC preparation is not unexpected, owing to the fraction of CSA in the material (SIGMA lot BCCC4066 used in these studies contained ~40% CSA), as often occurs with CS preparations.

Supplemental Figure S3: Full-length VAR2CSA recombinants bind specifically to decorin. ELISA plates were coated with decorin, CSC, HS or BSA and 5 $\mu\text{g}/\text{mL}$ of full length VAR2CSA (FCR3=blue, NF54=red, HB3=purple, 7G8=yellow, Malayan Camp=green, M920=orange, M200101=black) was bound to the different receptors. The amount of bound full-length VAR2CSA was detected by an HRP conjugated anti-his-tag.

Finally, the ELISA is not a good method to measure the affinity of a protein to its receptor. Surface plasmon resonance (SPR) experiments should be performed to measure the association strength of the different VAR2CSA to decorin and to provide the affinity constant K_D (K_{off}/K_{on}).

Response: We have measured the dissociation constants using Biacore for all 7 alleles (both glycosylated and deglycosylated variants) and amended the revised manuscript accordingly (Please see **Table 2** and **Supplemental Figure S2**). The relative binding strength predicted by ELISA was confirmed by K_D values of all alleles. Since the binding of VAR2CSA to CSA is complex and involves multiple domains, the stoichiometry is not 1:1, therefore kinetic analysis by Biacore is not possible (Srivastava A, et al. *Proc Natl Acad Sci U S A* 2010, 107:4884-4889 and Ma, R, et al. *Nature Microbiology* 2021). Instead, we analyzed the data as done previously for VAR2CSA (Srivastava A, et al. *Proc Natl Acad Sci U S A* 2010, 107:4884-4889) using steady state equilibrium analysis, and we measured dissociation constants consistent with previous values reported for the NF54 variant (Srivastava A, et al. *Proc Natl Acad Sci U S A* 2010, 107:4884-4889).

VAR2CSA allele	Glycosylated		Deglycosylated	
	K _D (nM)	SE(KD)	K _D (nM)	SE(KD)
7G8	35	1.3	27	0.6
NF54	59	1.1	87	0.28
HB3	84	10.8	65	0.49
M920	70	6.1	48	6.6
Camp	73	10.9	31	5.5
FCR3	860	106	520	53
M200101	130	0.16	61	5.4

Finally, as mentioned by the authors in the discussion, the expressed recombinant proteins are likely to be glycosylated since they didn't remove the N-glycosylation sites. Could these glycosylations affect the binding to CSA and the sera recognition of the different recombinant proteins? It could be interesting to perform deglycosylation experiments to assess the impact of glycosylation on the VAR2CSA binding properties.

Response: In order to assess the impact of glycosylation on VAR2CSA binding properties, we prepared deglycosylated variants of all 7 alleles by enzymatic treatment (Please see **Supplemental Figure S4**) for further studies. First, we demonstrated that removal of glycans from full-length VAR2CSA modestly increases the binding affinity of CSA (decorin) for all tested alleles except NF54, for which affinity was slightly decreased (**Table 2**). Second, we assessed the impact of glycosylation on antibody binding to VAR2CSA by reacting plasma IgG from 150 pregnant women to both glycosylated and deglycosylated forms of full-length VAR2CSA (**Figure 4 and Supplemental Figures S5, S6, S7, S8**). Our data indicate for all alleles, the deglycosylated form is less well recognized by plasma IgG compared to the glycosylated form (**Figure 4 and Supplemental Figures S5, S6, S7, S8**). This result is surprising because glycosylation modifications could be expected to decrease recognition by shielding epitopes that are exposed in the native protein. One possible explanation is that glycan modifications decrease the dynamics of the local protein structure (Rudd PM, Joao HC, Coghil E, Fiten P, Saunders MR, Opendakker G, Dwek RA (1994) *Biochemistry* 33:17–22; and Benoit I, Asther M, Sulzenbacher G, Record E, Marmuse L, Parsiegla G, Gimbert I, Asther M, Bignon C (2006). *FEBS Lett* 580:5815–5821). This decrease in local fluctuations may in turn preserve the local structure of the epitope and thereby enhance antibody binding.

Reviewer #2 (Remarks to the Author):

In the manuscript "Allelic variants of full-length VAR2CSA, the placental malaria vaccine candidate, differ in antigenicity and receptor binding affinity" Renn et. al., describes the production of 7 full length VAR2CSA recombinant proteins. In 2010 Srivastava published structural and biochemical data on the 3D7 full length VAR2CSA and Khunrae on the FV2

genotype. The present study extends the analyses to 5 more genotypes. In general the study is not very thorough and adds only little to the understanding of the structural features of VAR2CSA and pathology of this type of malaria.

Specific comments:

1) Figure 1: The manuscript presents a phylogenetic tree over a panel of full length VAR2CSA sequences. It is unclear how the tree is built, i.e. what type of tree is it. But more importantly with the block structure variation seen in PfEMP1 sequences it is meaningless to make a phylogenetic tree over the entire sequence. If the authors want to pursue this in a ligand binding context, they could have built a more reliable tree over the ligand binding region in DBL2 rather than the entire sequence.

Response: We have modified the tree format (as shown below, left image) in the revised manuscript to match a traditional phylogenetic tree layout. The thrust of this manuscript is expression and characterization of full-length VAR2CSA proteins, therefore, we constructed a tree over the entire protein sequence to guide us in our selection of alleles from distinct branches. As suggested by the reviewer, we constructed a tree using the larger ligand binding region of ID1-DBL2X boundaries (**below, right image**, Patel, J, et al, *Sci. Rep* 2017) and again showed that the alleles studied here represent several distinct phylogenetic branches. While the distribution differs somewhat from that seen in analysis of the full-length VAR2CSA, it still supports the selection of isolates expressed in this manuscript. Further, the recent report of the structure of VAR2CSA bound to CSA (Ma, R, et al. *Nature Microbiology* 2021) indicates that the binding interactions involve multiple domains including NTS, DBL1X, DBL2X and DBL4ε. If the reviewer and editor believe including the tree built from the ID1-DBL2X would benefit the readers, we will include it in the manuscript.

2) *Figure 2: The authors state that they have “high purity protein”. A silver stain or HPLC would reveal high level of impurities. Even with the current coomassie stain it is evident that quite a few of the proteins (7G8, MC, M0101) have a significant amount of impurities. A conservative estimate would be that for example 7G8 is only 50% pure, this makes comparison of yields meaningless when level of impurities differ that much. And as no SDS page is provided for Figure 2B it is not possible to evaluate comparative yields. As the proteins are only purified in a one step IMAC procedure, one would expect many impurities, and makes OD measurements of concentrations unreliable. Preferably the proteins should have been purified using a two step procedure (IMAC followed by IEX or SE). More worrying is the fact that not all the proteins change migration pattern after reduction, clearly one would expect a slower migration of these highly disulfide bonded proteins. This questions the quality of the proteins that doesn't follow this pattern. This finding is mentioned by the authors in the manuscript.*

Response: The purification included not only IMAC purification, but also concentration and diafiltration with a 100 kDa cutoff membrane. The manuscript is now updated for clarity.

In addition, we have now included a size exclusion chromatography step using an S6 column (please see **Figure 2B**). This column had a cutoff limit of 5,000 kDa so that full-length VAR2CSA could enter. The material showed high purity indicated by a single peak. We also increased the amount of protein loaded on the SDS-PAGE from 1 µg to 10 µg, and found the lanes were free of contaminating bands. Functional characterization of the recombinant proteins demonstrated that they are folded, have activity and are quite stable, suggesting that the different changes in migration pattern for some alleles reflects intrinsic differences between each protein, and are not due to quality of the expressed proteins.

3) *Figure 3A: A two point measure using CD doesn't make much sense. It would be standard to show a full melting curve to see any differences among the proteins.*

Response: We collected full melting curves for all 7 alleles (please see **Figure 3** below) and extracted T_m values for each allele (**Figure 3D**). Our results show high thermal stability with melting temperatures varying from 70°C-75°C, with NF54 being the most stable and M920 the least.

Figure 3: Biophysical characterization of full length VAR2CSA using circular dichroism spectroscopy. Far-UV wavelength scans of all full-length VAR2CSA alleles at A) 20°C and B) 95°C and thermal stability of 7 alleles, T_m range from 70°C-75°C.

Figure 3: Biophysical characterization of full length VAR2CSA using circular dichroism spectroscopy.

Far-UV CD wavelength scan of 0.2 mg/mL full-length VAR2CSA (FCR3=blue, NF54=red, HB3=purple, 7G8=yellow, Malayan Camp=green, M920=orange, M200101=black) at A) 20°C and B) 95°C, showing the loss of α -helical structure upon thermal denaturation. C) Thermal melt of all the 7 alleles monitoring the loss of CD signal at 200 nm and plotting the fraction unfolded vs. temperature. D) The T_m values of all 7 alleles obtained by the fit. NF54 is statistically different. Error is SEM based on 3 replicates.

4) Figure 3C. I assume that the assay is done with a titration of the protein and a titration of decorin as stated on the legend? One of the main point in the manuscript is that the different VAR2CSA proteins have different binding phenotypes, as described in Fig3C. This is a very superficial analyses, and seemingly several of the proteins have a flat low curve indicating unspecific binding. The assay lacks controls for specificity and to report fine differences I would suggest to use label free kinetic analyses (ITC or Biacore).

Response: We thank the reviewer for this suggestion and pointing out this typo on the x-axis, this mistake has been corrected. We have now measured the binding affinity of the different full-length VAR2CSA to CSA using the Biacore (please see response #4 to Reviewer 1 for more details). We have also determined the specificity of the VAR2CSA recombinants binding to CSA compared to CSC and HS. MSP1 protein was used here as a negative control and showed no binding to the different tested receptors (please see response #3 response to Reviewer 1 for more details).

5) Figure 5. With limited efforts the authors could have provided a proper serological

analyses. Using a single concentration of a pooled sample is meaningless. The authors have previously excelled in serological analyses of PfEMP1 proteins and would have sera ready available. At least end point titrations and subtypes could have done with the pools. And finally I wonder why the authors have not immunized with the proteins and shown the capacity of the immunogens to inhibit CSA binding of different genotypes of VAR2CSA binding parasites.

Response: In this revised version of the manuscript, we assessed plasma samples from 150 randomly selected pregnant or parturient Malian women for their levels of IgG binding to the different full-length VAR2CSA proteins. Our data confirm that antibody levels against VAR2CSA increase as a function of gravidity (**Figure 4 and Supplemental Figures S5 and S6**). No gravidity-related difference was seen in the plasma IgG reactivity to the control antigen MSP1. We also assayed the same set of samples with deglycosylated forms of the different VAR2CSA proteins, and demonstrated overall that antibodies bound at higher levels to the glycosylated compared to the deglycosylated forms of VAR2CSA (**Figure 4 and Supplemental Figure S5**), suggesting removal of glycosylation reduced antibody reactivity to VAR2CSA (**Figure 4 and Supplemental Figure S6**). This is now addressed in the Results (**lines 275-282**) and Discussion (**lines 378-386**).

We thank the reviewer for her/his suggestion to immunize animals and show binding inhibition activity of the recombinant proteins. These studies are a focus of future work.

Figure 4: Full-length VAR2CSA recombinants are recognized by naturally-acquired antibodies in gravidity and glycosylated-dependent manner.

ELISA plates were coated with full-length VAR2CSA. 150 samples from malaria-exposed multigravid (MG), secundigravid (SG) and primigravid (PG) women were diluted 1:1000 and measured for reactivity to all full-length VAR2CSA proteins. The mean from 50 samples at each gravidity is plotted with error bars representing the standard error of the mean. For each allele, the glycosylated protein is a solid line and the deglycosylated is dashed. Statistically significant differences between the values of the glycosylated and deglycosylated are indicated with p values <0.05=*, <0.01=**, <0.001=***, and <0.0001=****.

Reviewer #3 (Remarks to the Author):

The manuscript by Renn et al., contain important information that help us to understand placental malaria and progress in the design of protective vaccines to the vulnerable populations. They show high yield expression and purification of recombinant full length extracellular VAR2CSA antigen which produce 7 alleles. They compare structural, functional and antigenic characteristics and show that they vary in their binding to CSA. The rationale for focusing on the 7 fragments is well stated and the work helps move the malaria field forward.

1. Authors report some inconsistencies in the NF54 yield from 2 different runs (6 and 17 mg/mL), which they hypothesize that may be due to passage numbers and cell count, I think performing this experiment to confirm that it is true would make the manuscript much stronger.

Response: We have produced another batch of NF54 VAR2CSA using Expi293+pHLSEC, which yielded 13 mg/L (see **Table 2**), supporting the conclusion that production of the NF54 allele was improved when both Expi293 and pHLSEC were used for expression.

2. It would be better to classify the serum reactivity of the recombinants based on trimester of the pregnancy in the primigravid and multigravid groups and if this should show differences between the two groups as indicated in figure 4.

Response: In our revised manuscript we report data on individuals rather than pooled sera (please see response to reviewer 2 point 5 and new **Figure 4** and **Supplemental Figures S5 and S6**). The primigravidae, secundigravidae, and multigravidae groups who provided the sera did not differ significantly by trimester. As the reviewer implies, seroreactivity does increase by trimester, and we show the data stratified by trimester in the SI (**Supplemental Figure S7**). Using a linear regression model with both gravidity and trimester in the model both variables are highly significant (p-values: 2×10^{-16} for gravidity and 2×10^{-8} for trimester). However, the magnitude of the effect of gravidity is roughly 4-fold higher than trimester (+23.8 for gravidity and +6.8 for trimester). The new data and revised analysis support our original conclusion that immune serum reactivity to full-length VAR2CSA recombinants increases with gravidity.

Supplemental Figure S7: Recognition of full-length VAR2CSA recombinants by naturally-acquired antibodies increases with gestation age.

Normalized ELISA units were separated into different bins of gestational age. The bins were defined as 1st ≤13 weeks, 2nd 14-26 weeks and 3rd ≥27 weeks. For each allele, the glycosylated protein is a solid line and the deglycosylated is dashed. Comparisons of the effect of deglycosylation were performed using linear regression and 2-sample T-test. Statistically significant differences between the values of the glycosylated and deglycosylated are indicated with p values <0.05=*, <0.01=**, <0.001=***, and <0.0001=****.

3. There is no indication of the kind of statistics that were performed for data that show error bars.

Response: We have updated all figures and legends accordingly.

4. While the authors state that US controls were used, it is not clear in Fig 4 the data showing this controls or how they were incorporated in the analysis.

Response: We have now clarified this in the revised manuscript. A pool of US sera was used a negative control to normalize ELISA reactivity. In addition, 36 individual US naïve serum samples were also assayed. The Methods section, as well as figures and legends, have been modified accordingly.

Reviewers' comments:

Reviewer #1 (Remarks to the Author):

The revised version of the manuscript has improved considerably. This is a very interesting and well performed study that will be of interest for the community and for placental malaria vaccine development. I have no further editing request.

B. Gamain

Reviewer #2 (Remarks to the Author):

Thank you for the revised manuscript. It has been greatly improved and most of my comments has been addressed. I have a few comments (Two major and some minor) to the new data that should be addressed.

Major points:

1) The authors provide new data on de-glycosylated proteins. The only analyses of the deglyc proteins are SEC, please provide as a minimum also SDS page of the deglycosylated proteins, maybe that can resolve the issue around the strange migration patterns.

2) The authors test specificity of the binding in ELISA against Decorin, CSC, and "heparin sulfate" It just says "Sigma" so it's impossible to know what the reagents are. I assume that its heparan sulfate proteoglycan. And I assume that CSC is shark cartilage chondroitin sulfate with no protein core. In any case these reagents (and in particular CSC and CSPG/decorin) will coat the ELISA plate very differently (sugar coating vs protein coating) and it thus does not make sense to test in a direct binding assay, unless a positive control is used to assess coating, or some other way to assess if the sugar has been coated. One way to address specificity would be to coat with the CSPG and then inhibit in a titration with all the other reagents. This needs to be addressed.

Minor points.

1) " Affinity measurement by Biacore demonstrated a slight decrease in binding affinity of the deglycosylated protein for 6 of the 7 alleles, suggesting enhanced binding to CSA (Table 2)." Its not a decrease in affinity, but a decrease in Kd and an increase in affinity.

2) "and suggest a role for the C-terminal flexible flanking arm to mask epitope protective antibody epitopes as an additional immunoevasion strategy" There are no data in this manuscript to support this idea.

3) "However, the SDS-PAGE analysis highlighted that alleles of VAR2CSA recombinants differed in migratory patterns in non-reduced or reduced conditions, suggesting differing tertiary structure between alleles" I am still very puzzled about the SDS page migration pattern of the different proteins. It doesn't really make sense. And I do think it deserves a comment in the manuscript rather than just stating they "differed".

4) Figure 4 and others: It doesn't make sense to connect the means between the different population groups as they are fully independent, and there are no values on the X axis in between the M, S and P. The correct presentation would be box plots or something similar. Further the abbreviations used in the legend are different from the ones in the figure.

Reviewer #3 (Remarks to the Author):

The authors have sufficiently addressed the concerns that I had in the first draft of this manuscript.

Reviewer #1 (Remarks to the Author):

The revised version of the manuscript has improved considerably. This is a very interesting and well performed study that will be of interest for the community and for placental malaria vaccine development. I have no further editing request.

B. Gamain

Response: We thank the reviewer for the positive feedback and support of our manuscript.

Reviewer #2 (Remarks to the Author):

Thank you for the revised manuscript. It has been greatly improved and most of my comments has been addressed. I have a few comments (Two major and some minor) to the new data that should be addressed.

Major points:

1) The authors provide new data on de-glycosylated proteins. The only analyses of the deglyc proteins are SEC, please provide as a minimum also SDS page of the deglycosylated proteins, maybe that can resolve the issue around the strange migration patterns.

Response: In this updated manuscript, we have included the SDS-PAGE analysis of the deglycosylated proteins (**Supplemental Figure S4, B**) with description in **lines 274-283**.

Supplemental Figure S4 : Deglycosylation of full-length VAR2CSA by size-exclusion chromatography.

Supplemental Figure S4: Deglycosylated full-length VAR2CSA analyzed by size-exclusion chromatography and SDS-PAGE.

Size exclusion analysis of all 7 full-length VAR2CSA proteins ran on a Superose 6 column. Each trace is in a different color (FCR3=blue, NF54=red, HB3=purple, 7G8=yellow, M. Camp=green, M920=orange, M200101=black). All proteins eluted around 15 mL (void volume of the column is 8 mL) compared to the glycosylated variants that eluted around 14 mL (**Figure 2B**). Deglycosylated VAR2CSA recombinants were analyzed in SDS-PAGE gels. Ten µg of purified full-length VAR2CSA FCR3 (lane 1 and 9), NF54 (lane 2 and 10), 7G8 (lane 3 and 11), HB3 (lane 4 and 12) M. Camp (lane 5 and 13), M920 (lane 6 and 14), and 2 µg of M200101 (lane 7 and 15) were loaded on a 3–8% tris-acetate gel (with or without 100 mM DTT) and Coomassie stained. Lane 8 is the molecular weight marker (kDa).

2) The authors test specificity of the binding in ELISA against Decorin, CSC, and “heparin sulfate” It just says “Sigma” so it’s impossible to know what the reagents are. I assume that its heparan sulfate proteoglycan. And I assume that CSC is shark cartilage chondroitin sulfate with no protein core. In any case these reagents (and in particular CSC and CSPG/decorin) will coat the ELISA plate very differently (sugar coating vs protein coating) and it thus does not make sense to test in a direct binding assay, unless a positive control is used to assess coating, or some other way to assess if the sugar has been coated. One way to address specificity would be to coat with the CSPG and then inhibit in a titration with all the other reagents. This needs to be addressed.

Response: We thank the reviewer for the comments and have included catalog numbers for the reagents used in the specificity assay (please see **line 645**). To test for coating efficiency of decorin and CSC, we ran an ELISA using a monoclonal antibody that recognizes chondroitin sulfates (both CSA and CSC). Under the same plating conditions used in the specificity ELISA, we detected comparable signals to both decorin and CSC (please see below **Supplemental Figure S3, C**) suggesting that the density is the same and not influencing our results. We would also like to highlight that we coated with 10-fold more CSC to compensate for reduced protein core in CSC. Additionally, using an anti-heparin sulfate antibody to test coating of HS, we demonstrated that HS does not coat the plate sufficiently and therefore we have removed the data from the manuscript. We believe the comparison of decorin, CSC, and BSA comparison is enough to establish specificity to chondroitin 4-sulfate.

Supplemental Figure S3: Assessing binding specificity of VAR2CSA to CSA.

Supplemental Figure S3: Full-length VAR2CSA recombinants bind specifically to decorin.

ELISA plates were coated with decorin, CSC, or BSA and 5 µg/mL of full length VAR2CSA (FCR3=blue, NF54=red, HB3=purple, 7G8=yellow, M. Camp=green, M920=orange, M200101=black) was bound to the different receptors. Panel A shows the glycosylated VAR2CSA, and panel B shows the deglycosylated VAR2CSA. The amount of bound full-length VAR2CSA was detected by an HRP conjugated anti-his-tag. To test the density of decorin vs. CSC ELISA plates were coated with either decorin or CSC and the amount of chondroitin sulfate was measured by using 2 µg/mL of anti-chondroitin sulfate antibody that recognized both CSA and CSC.

Minor points.

1) "Affinity measurement by Biacore demonstrated a slight decrease in binding affinity of the deglycosylated protein for 6 of the 7 alleles, suggesting enhanced binding to CSA (Table 2)." Its not a decrease in affinity, but a decrease in Kd and an increase in affinity.

Response: We have updated this text to fix this error, please see **line 284**.

2) "and suggest a role for the C-terminal flexible flanking arm to mask epitope protective antibody epitopes as an additional immunoevasion strategy" There are no data in this manuscript to support this idea.

Response: We have removed this sentence from the text.

3) "However, the SDS-PAGE analysis highlighted that alleles of VAR2CSA recombinants differed in migratory patterns in non-reduced or reduced conditions, suggesting differing tertiary structure between alleles" I am still very puzzled about the SDS page migration pattern of the different proteins. It doesn't really make sense. And I do think it deserves a

comment in the manuscript rather than just stating they “differed”.

Response: Removal of the glycosylation modifications did not explain the puzzling reduction migration pattern of 3 of the recombinant VAR2CA alleles. We have added additional text in **lines 274-283** and **350-353** adding further discussion on the matter:

“Data presented in this study do not rule out that these atypical migration patterns could arise from incorrect folding or disulfide bond formation of the recombinant protein, however we believe this is unlikely since all three alleles display strong binding to CSA suggesting native function.”

4) Figure 4 and others: It doesn't make sense to connect the means between the different population groups as they are fully independent, and there are no values on the X axis in between the M, S and P. The correct presentation would be box plots or something similar. Further the abbreviations used in the legend are different from the ones in the figure.

Response: We have changed Figures 4 and 5 to box plots. Please see below the new figures.

Figure 4: Full-length VAR2CSA recombinants are recognized by naturally-acquired antibodies in gravidity- and glycosylation-dependent manner.

Figure 5: Recognition of full-length VAR2CSA recombinants by naturally-acquired antibodies increases with trimester.

Reviewer #3 (Remarks to the Author):

The authors have sufficiently addressed the concerns that I had in the first draft of this manuscript.

Response: We thank the reviewer for the positive feedback and support of our manuscript.

REVIEWERS' COMMENTS:

Reviewer #2 (Remarks to the Author):

The authors have addressed my key comments, suggestions and concerns, and I have no further comments.